# A Decade of Monitoring Primary Healthcare Experiences through the Lens of Inequality

**DOI:** 10.3390/healthcare12181833

**Published:** 2024-09-13

**Authors:** M. Isabel Pasarín, Maica Rodríguez-Sanz, Silvina Berra, Carme Borrell, Kátia B. Rocha

**Affiliations:** 1Agència de Salut Pública de Barcelona, 08023 Barcelona, Spain; mrodri@aspb.cat (M.R.-S.); cborrell@aspb.cat (C.B.); 2CIBER de Epidemiología y Salud Pública (CIBERESP), 28029 Madrid, Spain; 3Institut de Recerca Sant Pau (IR Sant Pau), 08041 Barcelona, Spain; 4Departament de Ciències Experimentals i de la Salut, Universitat Pompeu Fabra, 08003 Barcelona, Spain; 5Escuela de Salud Pública, Facultad de Ciencias Médicas, Universidad Nacional de Córdoba, Córdoba 5000, Argentina; sberra@unc.edu.ar; 6Centro de Investigaciones y Estudios sobre Cultura y Sociedad, Consejo Nacional de Investigaciones Científicas y Técnicas, y Universidad Nacional de Córdoba, Córdoba 5000, Argentina; 7School of Health and Life Sciences, Pontifical Catholic University of Rio Grande do Sul (PUCRS), Porto Alegre 90619-900, Brazil; katiabonesrocha@gmail.com; 8Departamento de Psicología Social y Metodología. Facultad de Psicología, Universidad Autónoma de Madrid (UAM), 28049 Madrid, Spain

**Keywords:** primary care, health inequalities, public health, access to care, health information

## Abstract

Background: Health care is not exempt from harboring social inequalities, including in those countries with a universal public system. The objective was to ascertain whether the population’s assessment of primary care (PC) changed between 2006 and 2016, the decade that included the economic crisis of 2008, and also if it exhibited patterns of social inequality in Barcelona (Spain). Methods: This was a cross-sectional study using Barcelona Health Surveys 2006 and 2016. Samples (4027 and 3082 respectively) comprised residents in Barcelona, over 15 years old. Dependent variable: Primary Care (PC) index. Independent variables: age, social class, and birthplace. Analyses included means and percentiles of PC index, and Somers’ D test to compare the distribution of the groups. Results: Comparing 2016 with 2006, the distribution of the PC index remained in women (median of 73.3) and improved in men (from 70 to 73.3). By social class, the pattern of inequality observed in 2006 in men with perceived poor health status disappeared in 2016. Inequalities according to birthplace persisted in women, regardless of perceived health status, but disappeared in men. Conclusions: In the 10 years between which the global economic crisis occurred, the assessment of PC did not worsen, and it did improve for men, but the study points to the need for more focus on people born abroad.

## 1. Introduction

Reducing health inequalities and enhancing the quality of healthcare remain significant challenges for all countries, even for those with universal health coverage. Strengthening primary care (PC) and expanding coverage have frequently been linked to reducing health or mortality inequalities [1], but few studies have addressed correlates between PC attributes and health equity [2]. The lack of empirical works may be a consequence of the weak focus on equity in frameworks for the improvement of PC quality [3]. Yet, most healthcare research does not stratify the population into social, economic, or geographic indicators to spot inequalities in the healthcare indicators across these axes of inequality [4].

In 1986, Spain adopted the model of universal health service coverage through the General Health Law. And a Primary Health Care (PHC) reform strategy was implemented between 1996 and 2003, in order to give this level of care a key role in the universalization and improvement of the quality of care for the entire population.

Barcelona is the second largest city in Spain, with more than 1.6 million inhabitants. Health surveys enabling the evaluation of policy impacts aimed at achieving health equity have been a primary strategy for monitoring inequalities in the city since 1983 [5,6]. The 2006 Health Survey added for the first time a short module to assess the population’s experiences with PC adapted for its implementation in Spain and psychometrically evaluated [7,8,9]. This assessment revealed satisfactory results, particularly among older individuals, those with more frequent doctor visits, and those with consistent provider relationships. Continuity in doctor-patient relationships over time appeared to enhance experience. Inequalities per se were not found according to social class, but a better experience was reported among those with private sector doctors and women with dual health coverage (public and private) [10]. Similarly, among children and adolescents, higher ratings were noted among those with dual health coverage and native-born parents compared to immigrants [11]. The fact the foreign-born population increased from 7.2% in 2000 to 17.5% in 2006 and 22.5% in 2016 [12] underscores the need to analyze immigration’s impact on inequalities.

The global economic downturn that commenced in 2008 led to targeted measures resulting in budget cuts across the Spanish healthcare system, impacting PC as well [13]. As in other countries, there were health austerity measures, including the reduction of public health spending, limitation of access to the public health system for certain groups, such as undocumented immigrants, and new co-payment measures for out-of-hospital pharmacies (pensioners, who until then had free medications as outpatients, had to pay 10%, with a few exceptions) [14]. As PC is the service that provides the greatest coverage of the population’s health needs, it is advisable to study whether a crisis affecting the health system as a whole left an impact once it was over. That is why this study proposes to explore whether the population’s assessment of this first level of care changed, and if changes are presented, whether they follow a pattern of inequality. Given that the health austerity measures in Spain were in effect until 2014 [15], we took advantage of the health surveys of the years 2006 and 2016, the whole austerity period of study having occurred between them. 

It is worth mentioning that a review of the Spanish health system [16] highlighted the central importance of PHC, and despite budget constraints, overall satisfaction with PC remained stable. However, only a third of those seeking care reported same-day appointments. Notably, the Health Barometer in 2012 showed an increase in the percentage of individuals perceiving a decline in service quality compared to previous years, yet satisfaction with PC and certain aspects such as proximity, accessibility, provider interaction, etc., remained steady [13]. Evaluations of PC between 2008 and 2018 in various Spanish Autonomous Communities, including Catalonia [17], exhibited stability, with scores above 86 out of 100. Conversely, a trend analysis of Spanish Health Surveys from 2001 to 2017 revealed a halt in the increase of PC utilization after 2011. Furthermore, the probability of using PC services increased among socioeconomically disadvantaged groups based on social class and education level, while the opposite trend was observed for specialized care [18], aligning with the established inequality pattern in specialized care utilization [19].

These discrepancies lend more relevance to the possibility of using indicators based on a model of PC’s expected attributes rather than subjective assessments of quality. Moreover, the Health Surveys afford the opportunity to monitor the PC experiences a decade after, delving. 

This paper delves into the pre- and post-economic crisis landscape, examining the trends of PC experiences, as well as changes in social inequalities in PC experiences. The equity hypothesis implies that utilization experiences should be equitable or better among groups with higher needs, independent of their ability to pay. Specifically, disadvantaged social class groups or migrants should not report worse PC experiences when they share similar health statuses. Furthermore, it is to be expected that older individuals and those with poorer health, having greater needs, would use more services without reporting worse PC experiences. Similarly, people with a closer relationship with PC should also show better PC experiences.

This study’s aims are to ascertain whether the population’s assessment of PC changes between 2006 and 2016, the decade that included the economic crisis of 2008 and to establish whether patterns of social inequality among women and/or men were exhibited according to age, social class, and birthplace and considering self-perceived health status, as well as the assessment of PC according to variables related to the reference doctor, in Barcelona (Spain) during 2006 and 2016.

## 2. Materials and Methods

A cross-sectional epidemiological study was conducted at two time points, 2006 and 2016, involving two separate samples of the population aged over 15 years in the city of Barcelona. Data were drawn from the Barcelona Health Surveys of 2006 [20,21] and 2016 [22,23]. These are population-based health interview surveys designed to collect data on health, health-related behaviors, health service utilization, and socio-demographic information from non-institutionalized individuals. Trained interviewers conducted face-to-face interviews in participants’ homes. The sample size for individuals aged 15 and above was 5398 in 2006 and 3514 in 2016. 

The final samples analyzed consisted of 4027 individuals in 2006 and 3082 in 2016. For this study, participants aged 15 and older who reported having a general practitioner, had visited a specialist doctor, and had answered at least 5 out of 9 questions in the PC rating index were included.

The PC assessment was conducted using the Primary Care Assessment Tool (PCAT)-based PC index, tailored specifically for population-based health surveys [9,10]. This index consists of a small set of questions covering various attributes of expected PC, with one item related to mental health comprehensiveness excluded due to modification done in 2016 [24]. Responses to index items were recorded on a Likert scale (1: “No, not at all”; 2: “Probably not”; 3: “Probably yes”; 4: “Yes, definitely”). In this study, the response option “don’t know or don’t remember” was replaced with the value zero, acknowledging ignorance as a potential access barrier to services [9]. The PC index score was calculated as the sum of the 9 items, yielding a possible score between 5 (given the minimum of 5 answered items) and 36. For ease of interpretation, the score was transformed to a 0–100 scale (score = 100 × [sum − 5]/[36 − 5]), with higher scores indicating a more positive PC experience.

The independent variables for analyzing inequalities included sex, age groups (15–44 years, 45–64 years, and ≥65 years), place of birth (Spain or abroad), and occupational social class (SC) categorized into non-manual workers (SC I, II, and III, the more privileged) and manual workers (SC IV and V, the less privileged) [25].

Other independent variables analyzed were type of general practitioner (public or private), duration of relationship with the same general practitioner (up to 2 years or 3 years or more), and number of visits to the general practitioner in the past year (none or 1 or more).

The analysis began by describing the distribution of the independent variables for each study year (2006 and 2016) by sex and including comparisons between 2006 and 2016 using the Fisher Exact test (Table 1). 

Subsequently, a descriptive analysis was performed for each of the nine PCAT items (Table 2), using both categorical (proportions %) and continuous measures (means and standard errors, median and interquartile range, IQR, that is, the interval between percentiles 25th and 75th), and including *p*-values from Somers’ D test to study trends in the distribution of each item between 2006 and 2016.

Somers’ D test provides information about the significance of differences in the distribution of a variable between groups based on rank statistics, being useful to compare non-normal distribution by groups when sample weighting is needed [26]. 

Then, trends in the distribution of the PC index between 2006 and 2016 were described and compared using Somers’ D measures (Table 3). Moreover, these comparisons between years were stratified by categories of independent variables, and on the other hand, comparisons between categories were stratified by year study (Table 4), all results separated by gender and perceived health status, grouping responses into good health (excellent, very good and good) and poor health (fair and poor) to detect inequalities that may affect specific gender and those with greater health needs or poor health [27,28,29,30].

Analyses were conducted using STATA 15.

## 3. Results

Table 1 provides an overview of the two studied samples. Both were made up of approximately 26% of individuals aged over 65, but some changes were observed in manual social classes with 49% in 2006 and 42% in 2016 (*p*-value < 0.001), and born outside Spain with 14% in 2006 and 22% in 2016 (*p*-value < 0.001). Over 70% had had their reference medical professional for 3 or more years. In 2006, 8% had not visited their medical professional in the last year, which increased to 20% in 2016 (*p*-value < 0.00). The percentage of those attended by a professional from the public system (Catalan Health Service) increased from 82% in 2006 to 88% in 2016 (*p*-value < 0.001).

Table 2 outlines each item that makes up the PC index and denotes its corresponding PC function. Most respondents visited the general practitioner (GP) or referral center. The response “definitely yes” decreased from 82% in 2006 to 76% in 2016, but a significant rise was observed in individuals who reported being able to access care at their center on the same day if they fell ill, increasing from 49% in 2006 to 69% in 2016. Telephone care accessibility registered relatively lower scores in both surveys. All three indicators of continuity of care by the same professional exhibited improvement between 2006 and 2016. However, the score related to the likelihood that this professional would be the one offering attention by telephone remained consistently low. Indicators of care coordination and cultural competence remained high and even improved.

The PC index (Table 3) showed a positive trend from 2006 to 2016. Although the median was 73 in both years, the mean and the 75th percentile increased slightly, and the distribution of the index changed between both years, leading to the *p*-value being significant. This improvement was particularly prominent in men, with a three-point increase in the median and a seven-point increase in the 75th percentile between the two surveys. 

Table 4 presents the distribution of the PC index (median along with 25th and 75th percentiles) for women and men, stratified by perceived health and different study variables. Across both genders and years, the PC index increased by age group. For men, there was an increase in the index across all age groups through the years (from 2006 to 2016).

Regarding social class, no pattern of inequality was observed for women. Among men in poor health, the gap between non-manual and manual social classes narrowed in 2016 from 10 points in 2006 (medians of 76.6 for men in non-manual and 66.7 for manual social classes) to 3.3 points in 2016 (80 and 76.7, respectively). This was due to the improvement of the PC index of men of manual social classes.

Inequalities based on country of birth existed in both women and men, with foreign-born individuals reporting a poorer evaluation of PC. It is worth mentioning that the PC index of foreign men increased between 2006 and 2016 disappearing inequalities in the last year. 

For both women and men, longer duration with the same professional correlated with a higher PC index score. Generally, having done 1 or more visits to the reference professional in the past year led to a better PC index score. Over a decade, the PC index decreased for women who had not participated in any visits.

Finally, both women and men had higher index scores when their reference doctor was in the private sector. In 2016, significant PC assessment improvements were observed among users of the public system among women who reported good self-perceived health and in men, regardless of health status.

## 4. Discussion

This study focuses on assessing the PC experience of the population in the city of Barcelona through health surveys conducted before (2006) and after (2016) a period of global economic crisis, which had significant impacts on living conditions and reduced public resources for the healthcare system. The analysis not only explores changes over this decade but also seeks to identify potential social inequalities in experiences with healthcare services.

The soundness of PC in Spain has garnered international recognition [31], yet there have also been calls for improvement [32]. In this study, the vast majority of the sample reported having a designated reference professional, a fundamental aspect of PC as a gateway to the healthcare system and a key component as the longitudinal attribute of person-centered care [33]. Most of these professionals or reference centers are within the public system [21,23], reflecting the extensive coverage of Spanish public services.

Average item scores exceeding 3 (on a scale of 1–4) or 66 (on a scale of 0–100) are considered satisfactory, and in the international context, the values observed here are high [34,35]. The differences between the two time points studied indicate that, in general, the upper quartile of PC index scores improved significantly. Conversely, the stability of the median value over time could be interpreted as a plateau, such as findings in Canada when comparing 2007 and 2016 [36]. Nevertheless, given the reductions in public budgets affecting the healthcare sector during the economic crisis [13,14,37,38], as pointed out by B López-Varcárcel and P Bárber, there was no irreversible deterioration, most likely due to the dedication of health personnel and good management [14]. The results observed in this study could be explained in part by PC’s efforts to maintain quality in interactions with the population.

Our item analysis demonstrates improvements in indicators related to first contact, continuity of care, coordination, and cultural competence, alongside weaknesses in telephone accessibility, which did not show improvement during the studied period. The enhancement of telephone care assessment occurred a few years ago and might be attributed to its prioritization and revision during the COVID-19 pandemic, leading to increased usage. The latest Barcelona health survey of 2021 indicates that the percentage of individuals aware of telephonic access to their PC center has increased, although 24% remain unaware of this option when centers are closed [39]. 

The better assessment of PC as age increases in women and in those who use more services is a pattern already seen in previous studies [10,35]. Interestingly, an improvement in men’s assessment was observed here, more pronounced in healthy older adults and in young men who reported poor perceived health, and the pattern of inequality by social class disappeared in the group of men with poor perceived health status, which was the only group with poor perceived health status in 2006.

This study was aimed at assessing PC with a focus on equity to address social health inequalities. Over the ten-year period studied, the samples reflect an increase in the proportion of immigrant populations and a higher percentage of manual workers and users of public services, likely influenced by the economic crisis. The results confirm inequalities based on place of birth in 2006, which disappeared among men in 2016 but persisted among women regardless of perceived health status. Inequalities based on social class narrowed during the study period, which is noteworthy in the context of the crisis. The improvement in the index among men and foreign-born individuals could be linked to increased utilization and a better understanding of services. During the economic crisis, they might have sought more PC due to increased unemployment-associated illness risk [13,40], and immigrants may have lived in Barcelona for an extended period. The significant influx of immigrants in the early 21st century, along with local policies aimed at lowering access barriers, might have influenced this improvement. Additionally, prior studies have shown that longer patient-doctor relationships and more frequent visits result in a higher assessment of PC [10].

This study implemented the most widely used instrument among the different models and tools developed to assess the quality of PC [34,41,42]. The PCAT was developed in the United States of America by Starfield [33], and has been implemented in countries across continents [34,43]. Other studies in Spain implemented more extensive versions of PCAT, with greater validity for achieving multidimensional information on PC experiences [8,44], but the 10-item version was more suitable for inclusion in health surveys and had high reliability, as was proved [9,24].

In Brazil, it has been widely applied to assess health system reforms, given the broad coverage of the unified public healthcare system. The 2019 national health survey utilizing PCAT with recent users found that PC assessment tends to be higher among those who use more services (women, older adults, and individuals with chronic illnesses). Differences in PC assessment based on sex, ethnicity/race, or income were not significant, but substantial differences were found based on provider type (Family Health Programme, traditional units, or private clinics) and across states [35,45]. Several studies in Asia explicitly aimed to explore socio-economic inequalities in PC with PCAT assessment and confirmed variations between higher income groups and those with private health coverage [46,47]. 

In another context, the PHC was evaluated through the lens of social inequalities and highlighted room for improvement. Australia achieved universal health insurance several decades ago, giving a fundamental role to PHC, but the gap in addressing equity has been persistent when analyzing service delivery to socio-economically disadvantaged populations, including indigenous people, and others [48]. The Swedish reform enabled the privatization of PC centers, which contributed to a small improvement in overall PHC performance, though simultaneously to an increase in avoidable hospitalizations and socioeconomic inequities [49]. Our results showed an increase in the percentage of citizens consulting public facilities, and it is possible to hypothesize that this trend has continued in recent years. In addition, the government has reinforced programs aimed at improving access to primary care for the population that has recently arrived in Barcelona.

This study has certain limitations. First, the study sample includes individuals who identified a general practitioner, visited specialized care and answered at least 5 of the 9 index items. As a result, the findings cannot be extrapolated to the entire population, as a portion of the sample was excluded (25% in 2006 and 12% in 2016). However, since the primary goal is to analyze care equity, this selection does not undermine the results, as those excluded likely represent healthier individuals who utilize healthcare services less. Moreover, the inclusion of all types of care, both public and private, mitigates this effect. Second, all foreign-born individuals are grouped together due to an inability to differentiate between those from low- and high-income countries. This limitation prevents the identification of inequalities within the foreign-born group due to the small numbers, making further disaggregation unfeasible. It is important to consider that large sample sizes can influence *p*-values; as a result, minor differences might show statistically significant p-values. Lastly, the cross-sectional design might introduce reverse causality bias, where previous service experiences could condition use and experiences measured in the study.

Despite these limitations, the use of population-based health surveys is a robust approach for evaluating health policies and services, especially when conducted periodically as in the case of Barcelona Health Surveys. Individual-reported health, service experiences, and socio-demographic characteristics provide valuable indicators for analyzing needs, health determinants, and inequalities, and for drawing conclusions regarding the evolution of PC from the point of view of equity.

## 5. Conclusions and Recommendations

In conclusion, this analysis of two health surveys in the city of Barcelona showed that in a decade marked by the economic crisis, which also affected the health sector, the scores for citizens’ evaluation of PC remained high, inequality gaps by social class were reduced, and, although decreasing, there are inequalities between natives and immigrants. In line with what López-Valcarcel and Bárber pointed out for the health system as a whole, i.e., its resistance to presenting great deterioration after the economic crisis, in this specific work on PC, it can also be deduced that this level of care and its wide use of the public system, together with the care of the professional sector, including its management, are public policies to be reinforced. The PC, with its population orientation as intended in a national health system environment, is the part of the health system that can best act for the equity of health care [50].

Monitoring through repeated health surveys makes it possible to identify the existence or not of inequalities in access to and use of health services, as well as to evaluate the strategies used to reduce inequalities and the impact of new policies in favor of equity. PC, as the gateway to services, the provider of the greatest burden of disease in the population, and the coordinator of care, has an important role to play in reducing health inequalities. The different crises that may affect the population, whether economic, social, or health (such as the recent COVID-19 pandemic), can have impacts on health and health care beyond the crisis situation, and the population-based health surveys that are repeated periodically, as is the case in Barcelona, can help to monitor their long-term impacts, always carried out through the lens of inequality and with an intersectional approach.

To implement all these recommendations, the budget of PHC should be increased [51] to have more human resources and better quality of jobs. Moreover, it is necessary to prioritize equity-oriented policies in health plans, intersectoral plans, and in the governance of PHC. 

## Figures and Tables

**Table 1 healthcare-12-01833-t001:** Description of the variables and comparison between the 2006 and 2016 study samples.

	Total				Women				Men			
	2006 (N = 4027)		2016 (N = 3082)		2006 (N = 2295)		2016 (N = 1685)		2006 (N = 1732)		2016 (N = 1397)	
	N	%	N	%	N	%	N	%	N	%	N	%
**Age**												
15–44	1824	45.3	1316	42.7	1000	43.6	675	40.0	824	47.6	641	45.9
45–64	1146	28.5	940	30.5	647	28.2	507	30.1	499	28.8	433	31.0
= >65	1057	26.2	826	26.8	648	28.2	503	29.9	409	23.6	323	23.1
*p-*value ^1^				*0.262*				*0.708*				*0.286*
**Social class**												
Non-manual	1948	48.4	1663	54.0	1058	46.1	909	54.0	889	51.3	754	54.0
Manual	1978	49.1	1304	42.3	1152	50.2	698	41.4	826	47.7	606	43.4
Not known	101	2.5	115	3.7	85	3.7	78	4.6	17	1.0	37	2.6
*p*-value ^1^				*<0.001*				*<0.001*				*0.072*
**Birthplace**												
Spain	3457	85.8	2393	77.7	1969	85.8	1296	76.9	1488	85.9	1097	78.5
Foreign	570	14.2	682	22.1	326	14.2	389	23.1	244	14.1	293	21.0
Not known			7	0. 2							7	0.5
*p*-value ^1^				*<0.001*				*<0.001*				*<0.001*
**Time with the same reference doctor**												
<3 years	1011	25.1	834	27.1	575	25.1	446	26.5	436	25.2	388	27.8
= >3years	2840	70.5	2239	72.6	1633	71.1	1232	73.1	1207	69.7	1007	72.1
Not known	176	4.4	9	0.3	87	3.8	7	0.4	89	5.1	2	0.1
*p-*value ^1^				*0.162*				*0.659*				*0.108*
**No. of visits to the reference doctor (in the last year)**												
None	313	7.8	621	20.1	134	5.8	295	17.5	179	10.3	326	23.3
1 or more	3709	92.1	2461	79.9	2158	94.0	1390	82.5	1551	89.6	1071	76.7
Not known	5	0.1			3	0.2			2	0.1		
*p-*value ^1^				*<0.001*				*<0.001*				*<0.001*
**Physician modality**												
Public ^2^	3293	81.8	2732	88.7	1873	81.6	1483	88.0	1420	82.0	1249	89.4
Private ^3^	733	18.2	349	11.3	421	18.4	201	11.9	311	18.0	148	10.6
Not known	1	<0.1	1	<0.1			1	0.1	1	<0.1		
*p-*value ^1^				*<0.001*				*<0.001*				*<0.001*

^1^ *p*-value from Fisher’s Exact test to compare 2006 and 2016 (excluding missings).^2^ Catalan Health Service. ^3^ Include private health service and mutual health insurance. *p*-values are indicated in italics so as not to confuse them with the % values in the same column.

**Table 2 healthcare-12-01833-t002:** Description of the nine items that make up the PC index using both measures categorical (proportions %) and continuous (mean, standard error, SE, median and interquartile range, IQR, and the comparison between 2006 and 2016 distributions (Somers’ D test).

Categorical							Continuous				
		0 Do Not Know	1 Definitely Not	2 Probably Not	3 Probably Yes	4 Definitely Yes	[0–4]				
		%	%	%	%	%	Mean	SE	Median	IRQ	*p* Value ^1^
**First contact**											
	**1. When you have new health problems, do you go to your doctor before going somewhere else?**										<0.001
2006		0.1	1.9	4.0	12.3	81.7	3.7	0.6	4	4–4	
2016		0.2	2.0	4.7	17.4	75.7	3.7	0.7	4	4–4	
	**2. When the office is open and you get sick, would someone from there see you the same day?**										<0.001
2006		5.6	10.1	9.9	25.1	49.3	3.0	1.2	3	2–4	
2016		4.5	2.8	5.2	18.6	68.9	3.4	1.0	4	3–4	
	**3. When the office is open, can you get advice quickly over the phone if you need to?**										0.345
2006		22.3	8.5	8.0	22.7	38.5	2.5	1.6	3	1–4	
2016		28.4	5.8	3.7	22.4	39.7	2.4	1.7	3	0–4	
	**4. When the office is closed, is there a phone number you can call when you get sick?**										0.030
2006		15.4	6.0	4.3	16.7	57.6	3.0	1.5	4	2–4	
2016		19.2	6.4	1.3	17.3	55.8	2.8	1.6	4	1–4	
**Continuity**											
	**5. When you go to see your doctor, do you see the same doctor or nurse each time?**										<0.001
2006		1.8	5.8	8.0	23.6	60.8	3.4	1.0	4	3–4	
2016		2.2	3.8	5.6	20.8	67.6	3.5	0.9	4	3–4	
	**6. If you have a question, can you call and talk to the doctor who knows you best?**										0.453
2006		25.0	15.9	12.6	19.6	26.9	2.1	1.6	2	1–4	
2016		34.2	9.4	4.1	17.6	34.8	2.1	1.7	3	0–4	
	**7. Does your doctor know what problems are most important to you?**										<0.001
2006		3.0	9.5	10.3	20.7	56.5	3.2	1.1	4	3–4	
2016		3.8	5.5	4.9	17.8	68.0	3.4	1.1	4	3–4	
**Coordination**											
	**8. After going to the specialist, did your doctor talk with you about what happened at the visit?**										<0.001
2006		6.8	14.5	8.9	19.7	50.1	2.9	1.3	4	2–4	
2016		7.3	12.5	5.8	16.5	57.9	3.1	1.3	4	2–4	
**Cultural competence**											
	**9. Would you recommend your doctor to a friend or relative?**										0.033
2006		4.2	7.5	5.7	20.8	61.8	3.3	1.1	4	3–4	
2016		7.8	4.7	3.8	18.4	65.3	3.3	1.2	4	3–4	

^1^ *p*-value from Somers’s D test to compare items’ distribution between 2006 and 2016.

**Table 3 healthcare-12-01833-t003:** Summary measures of PC index by year (mean and standard error, SE, median and interquartile range, IQR), and comparison between 2006 and 2016 distributions (*p*-value of Somers’ D test).

	N *	Mean	SE	Median	IQR
**Total**					
2006	4022	70.0	20.1	73.3	56.7–86.7
2016	3083	72.2	21.0	73.3	56.7–90.0
*p-*value ^1^	*<0.001*				
**Women**					
2006	2292	71.0	20.1	73.3	56.7–86.7
2016	1685	72.4	20.6	73.3	56.7–90.0
*p-*value ^1^	*0.042*				
**Men**					
2006	1730	68.6	20.1	70.0	53.3–83.3
2016	1397	72.0	21.4	73.3	56.7–90.0
*p-*value ^1^	*<0.001*				

^1^ *p*-value for trend, from Somers’ D test to compare PC index distribution between 2006 and 2016. * Number pf people. *p*-values are indicated in italics to differentiate them from the type of values in the same column.

**Table 4 healthcare-12-01833-t004:** Description of PC index distribution using median, interquartile range (IQR), and the comparison between groups (Somers’ D test), stratified by sex and perceived health status, both (1) between 2006 and 2016 in each category of independent variables and (2) between categories of independent variables in each period.

**Women**						
	**Good Health**			**Poor Health**		
	**2006**	**2016**	***p-*Value ^1^**	**2006**	**2016**	***p-*Value ^1^**
**Age**						
15–44	66.7 (50.0–83.3)	70.0 (50.0–86.7)	*0.092*	70.0 (60.0–86.7)	66.7 (53.3–83.3)	*0.172*
45–64	73.3 (56.7–86.7)	73.3 (56.7–93.3)	*0.077*	76.7 (63.3–86.7)	76.7 (63.3–90.0)	*0.206*
= >65	80.0 (70.0–90.0)	76.7 (66.7–93.3)	*0.698*	80.0 (63.3–90.0)	76.7 (63.3–90.0)	*0.889*
*p-*value ^2^	*<0.001*	*<0.001*		*0.005*	*0.023*	
**Social class**						
Non-manual	70.0 (56.7–86.7)	73.3 (53.3–90.0)	*0.064*	76.7 (63.3–90.0)	76.7 (63.3–93.3)	*0.694*
Manual	70.0 (56.7–86.7)	73.3 (56.7–90.0)	*0.104*	76.7 (60.0–86.7)	73.3 (60.0–86.7)	*0.624*
*p-*value ^2^	*0.542*	*0.493*		*0.063*	*0.561*	
**Birthplace**						
Spain	73.3 (56.7–86.7)	73.3 (56.7–90.0)	*0.037*	76.7 (63.3–90.0)	76.7 (63.3–90.0)	*0.333*
Foreign	60.0 (46.7–80.0)	70.0 (50.0–90.0)	*0.002*	70.0 (56.7–83.3)	70.0 (56.7–80.0)	*0.621*
*p-*value ^2^	*<0.001*	*0.032*		*0.022*	*<0.001*	
**Time with the same reference doctor**						
Up to 2 years	63.3 (50.0–80.0)	66.7 (50.0–83.3)	*0.382*	70.0 (56.7–86.7)	70.0 (56.7–83.3)	*0.871*
= > 3 years	73.3 (56.7–86.7)	73.3 (56.7–93.3)	*0.067*	80.0 (63.3–90.0)	76.7 (66.7–93.3)	*0.543*
*p-*value ^2^	*<0.001*	*<0.001*		*<0.001*	*<0.001*	
**No. of visits to the reference doctor (in the last year)**						
None	70.0 (50.0–83.3)	63.3 (46.7–83.3)	*0.401*	76.7 (66.7–96.7)	70.0 (60.0–80.0)	*0.202*
1 or more	73.3 (56.7–86.7)	73.3 (56.7–90.0)	*<0.001*	76.7 (63.3–90.0)	76.7 (60.0–90.0)	*0.476*
*p-*value ^2^	*0.348*	*<0.001*		*0.560*	*0.059*	
**Physician modality**						
Public ^1^	70.0 (53.3–83.3)	70.0 (53.3–86.7)	*<0.001*	73.3 (63.3–86.7)	73.3 (60.0–86.7)	*0.718*
Private ^2^	83.3 (66.7–93.3)	83.3 (66.7–96.7)	*0.703*	80.0 (66.7–90.0)	100.0 (73.3–100.0)	*0.111*
*p-*value ^2^	*<0.001*	*<0.001*		*0.008*	*0.017*	
**Men**						
	**Good Health**			**Poor Health**		
	**2006**	**2016**	***p*-Value ^1^**	**2006**	**2016**	***p-*Value ^1^**
**Age**						
15–44	66.7 (50.0–80.0)	70.0 (50.0–86.7)	*0.037*	63.3 (50.0–80.0)	80.0 (53.3–86.7)	*0.019*
45–64	70.0 (56.7–86.7)	73.3 (56.7–90.0)	*0.063*	70.0 (56.7–86.7)	76.7 (60.0–96.7)	*0.077*
= >65	76.7 (63.3–90.0)	83.3 (66.7–96.7)	*0.011*	76.7 (63.3–90.0)	80.0 (70.0–90.0)	*0.161*
*p-*value ^2^	*<0.001*	*<0.001*		*<0.001*	*0.210*	
**Social class**						
Non-manual	70.0 (53.3–83.3)	70.0 (53.3–90.0)	*0.048*	76.7 (63.3–90.0)	80.0 (70.0–90.0)	*0.214*
Manual	70.0 (53.3–80.0)	73.3 (56.7–90.0)	*<0.001*	66.7(56.7–83.3)	76.7 (60.0–90.0)	*0.008*
*p-*value ^2^	*0.211*	*0.422*		*0.001*	*0.145*	
**Birthplace**						
Spain	70.0 (56.7–86.7)	73.3 (56.7–90.0)	*0.005*	73.3 (60.0–90.0)	80.0 (66.7–90.0)	*0.013*
Foreign	60.0 (50.0–73.3)	70.0 (53.3–86.7)	*<0.001*	60.0 (40.0–73.3)	76.7 (53.3–86.7)	*0.001*
*p-*value ^2^	*<0.001*	*0.099*		*<0.001*	*0.290*	
**Time with the same reference doctor**						
Up to 2 years	60.0 (46.7–76.7)	70.0 (50.0–86.7)	*0.002*	63.3 (53.3–76.7)	76.7 (60.0–86.7)	*0.004*
= > 3 years	73.3 (56.7–86.7)	73.3 (56.7–90.0)	*0.057*	76.7 (60.0–90.0)	80.0 (66.7–90.0)	*0.059*
*p-*value ^2^	*<0.001*	*<0.001*		*<0.001*	*0.228*	
**No. of visits to the reference doctor (in the last year)**						
None	60.0 (43.3–73.3)	63.3 (46.7–83.3)	*0.055*	70.0 (60.0–86.7)	70.0 (53.3–86.7)	*0.545*
1 or more	70.0 (56.7–83.3)	73.3 (60.0–90.0)	*<0.001*	73.3 (60.0–86.7)	80.0 (66.7–90.0)	*<0.001*
*p*-value ^2^	*<0.001*	*<0.001*		*0.961*	*0.026*	
**Physician modality**						
Public ^1^	66.7 (50.0–80.0)	70.0 (53.3–86.7)	*<0.001*	70.0 (56.7–86.7)	76.7 (63.3–90.0)	*<0.001*
Private ^2^	83.3 (70.0–93.3)	83.3 (60.0–96.7)	*0.856*	83.3 (73.3–96.7)	86.7 (73.3–93.3)	*0.910*
*p-*value ^2^	*<0.001*	*0.002*		*0.007*	*0.286*	

^1^ *p*-value for trend from Somers’ D test to compare PC index distribution between 2006 and 2016. ^2^ *p*-value for patterns from Somers’ D test to compare PC index distribution between categories. *p*-values are indicated in italics to differentiate them from the type of values in the same column.

## Data Availability

The Barcelona Health Surveys forms part of the statistical actions of interest to the Generalitat de Catalunya and is included in the Annual Statistical Action Programme (PAAE) under the registration number: 05-03-24. It is anonymous and confidential, in accordance with Law 6/2007, of 17 July, which regulates the preparation and publication of surveys and opinion polls in Catalonia. On the other hand, the confidentiality of the data is guaranteed in accordance with Organic Law 3/2018 on the Protection of Personal Data and the guarantee of digital rights. Therefore, it is assured that the information obtained from the questionnaires will be used exclusively in the field of health. According to the regulation, access to the data is on direct request to the administration (Public Health Agency of Barcelona) upon request by accredited research groups, under the clause of non-transfer to third parties. Therefore, access to the data is not allowed without their consent and subrogated access is not possible. On legal grounds, data is only accessible upon request from the official administrative source to the address info@aspb.cat.

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
