# Peer review of "A Decade of Monitoring Primary Healthcare Experiences through the Lens of Inequality"

_healthcare, 2024, doi:10.3390/healthcare12181833_

Round 1
Reviewer 1 Report
Comments and Suggestions for Authors
Good paper. Overall well written with areas for improvement:
1) Check abbreviations are first spelt out where first mentioned. PHC (line 40)
2) Methods: Describe whats somer' D index is. Why difference in original sample and sample analyzed. no mention of what statistical tests used for comparisons.
3) Results. why no p values for comparisons in Table 1. It seems like the proportions for social class, birthplace and healthcare utilization were significantly different.
4) In Table 3 means and medians are presented. What tests were used for the comparisons between both years? The medians are the same for total and women but the p values are significant. How do you explain this?
5) Instead of Table 4. why no regression analyses comparing 2016 to 2006 with adjustment for confounders like age and birthplace. This may be more important than a table of 48 p values. Unadjusted and adjusted analyses would be much more informative and be easier to interpret.
6) The significant findings had apparently very small effect sizes. What is the real world significance on these changes? In samples with over 3,000 significant comparative differences are inevitable because of overpowered samples. The discussion should include the magnitude of the sig. findings and how meaningful they really are.
7) limitations: no mention of type 1 error inflation which is more likely with large samples
Author Response
Response to the Review 1
We are grateful for the detailed peer review of our work and the particularly timely feedback we have received regarding on our manuscript ‘A Decade of Monitoring Primary Healthcare Functions Through the Lens of Inequality’ [healthcare-3174086], submitted for the Special Issue ‘Perspectives on Primary and Community Healthcare’ of Heathcare. We are also very grateful to the editorial team that has accepted that our work may have some chance of being published in the journal, as well as to the interest shown in the good editorial work.
We have addressed all the comments made and tried to make all the changes suggested for accuracy and quality. In our responses (Re) to each comment, we indicate page (p.) and paragraph (para.) and or lines and modifications with change tracking. We mark the comments received in red and our responses in black and blue.
Thank you very much for the review, the positive assessment of the work as well as the suggestions and proposed improvements, as they have especially allowed us to improve the presentation of the analysis carried out and the results, in addition to the rest of the proposals made.
Comments and Suggestions for Authors
Good paper. Overall well written with areas for improvement:
1) Check abbreviations are first spelt out where first mentioned. PHC (line 40)
Re: The spelt was added and abbreviations in the entire manuscript were revised. Now in line 45.
2) Methods: Describe whats somer' D index is. Why difference in original sample and sample analyzed. no mention of what statistical tests used for comparisons.
Re: This information has been added to the Methods section. In this section, the analysis part has been better explained, including the Somme' D index (lines 146-166)
3) Results. why no p values for comparisons in Table 1. It seems like the proportions for social class, birthplace and healthcare utilization were significantly different.
Re: We have added the p values in the table (see the new table) and included some of them in the text.
4) In Table 3 means and medians are presented. What tests were used for the comparisons between both years? The medians are the same for total and women but the p values are significant. How do you explain this?
Re: P values compare the PC index distribution (not only the median). We have explained it in the Methods section and included in the title of Table 3. The p values have been corrected due to a previous error.
5) Instead of Table 4. why no regression analyses comparing 2016 to 2006 with adjustment for confounders like age and birthplace. This may be more important than a table of 48 p values. Unadjusted and adjusted analyses would be much more informative and be easier to interpret.
Re: Our interest was to show the pattern of inequalities in PC index in 2006 and 2016 and also if the increase in the PC index between 2006 and 2016 occurred stratifying for independent variables. The model suggested would not show these results, for this reason we think it is better to maintain the table. The explanation of the table has been simplified in order not to explain all p values.
6) The significant findings had apparently very small effect sizes. What is the real world significance on these changes? In samples with over 3,000 significant comparative differences are inevitable because of overpowered samples. The discussion should include the magnitude of the sig. findings and how meaningful they really are.
Re: Take into account that many p-values are non significant. As said above, we have simplified the text of table 4 and added a sentence in the limitations section.
7) limitations: no mention of type 1 error inflation which is more likely with large samples
Re: We have added a sentence in the limitations section (lines 360-365)
Reviewer 2 Report
Comments and Suggestions for Authors
please see the attachment

please see the attachment
Author Response
Response to the Review 2
We are grateful for the detailed peer review of our work and the particularly timely feedback we have received regarding on our manuscript ‘A Decade of Monitoring Primary Healthcare Functions Through the Lens of Inequality’ [healthcare-3174086], submitted for the Special Issue ‘Perspectives on Primary and Community Healthcare’ of Heathcare. We are also very grateful to the editorial team that has accepted that our work may have some chance of being published in the journal, as well as to the interest shown in the good editorial work.
We have addressed all the comments made and tried to make all the changes suggested for accuracy and quality. In our responses (Re) to each comment, we indicate page (p.) and paragraph (para.) and or lines and modifications with change tracking. We mark the comments received in red and our responses in black and blue.
REVIEWER 2
Thank you very much for the review, the suggestions and improvements that have been proposed. The English, in this new version, has been revised again by a native English teacher from the United States. The introduction has been the section that has been modified the most, in order to be able to meet the suggested proposals, better focusing the work and updating part of the bibliography. In the comments on each of the points indicated in the review, the changes made are specified.
Comments and Suggestions for Authors
Overall Assessment: The manuscript offers a comparative analysis of primary care (PC) assessments in Barcelona, Spain, during 2006 and 2016, focusing on social inequalities in healthcare access and utilization. The study is methodologically sound, utilizing cross-sectional data from the Barcelona Health Surveys, and provides valuable insights into the persistence and changes in social inequalities within the context of a universal healthcare system. The manuscript is relevant to the journal Healthcare as it aligns with the journal’s focus on public health, healthcare systems, and social determinants of health. However, significant revisions are necessary to improve the clarity, coherence, and scholarly rigor of the manuscript.
- Introduction: The introduction provides a broad overview of the importance of monitoring health inequalities, particularly in the context of primary care. However, the section could benefit from a more focused and concise articulation of the study's objectives and its significance. The current introduction touches on several points without clearly establishing the research question or the rationale for comparing 2006 and 2016. Additionally, the literature review embedded in the introduction could be expanded to include more recent studies on health inequalities and primary care assessments in other contexts. This would help position the study within the broader academic discourse and highlight its unique contributions.
Re: The introduction was modified to cover the issues suggested by the reviewer. The contextualization of primary care reform, assessment, and results in Spain was reorganized.
In the introduction, two paragraphs (lines 48-76) have been added, with two new bibliographic references (num- 14 and 15), which help to focus the research question based on the economic crisis of 2008 and specifically whether the population's assessment of their experiences with the first level of health care had changes between two years, 2006 and 2016. The health cuts made in Spain as a result of the economic crisis were not recovered until after 2014. Then, the rationale for comparing 2006 and 2016, and the need to focus on social inequalities has been more clearly established.
Although some updated references have been added to the introduction, the studies on primary care assessments in other contexts are commented on the discussion (p. 12, lines 342-349 is the place with new references).
- Literature Review: The literature review is somewhat fragmented and lacks a clear thematic structure. The manuscript would benefit from a more systematic review of the existing literature on primary care inequalities, particularly studies that have utilized similar methodologies or focused on similar populations. There is a need for a deeper engagement with the theoretical frameworks that underpin the study, particularly in relation to social determinants of health and health equity. The authors should also address any gaps in the literature that this study aims to fill, thereby strengthening the justification for the research.
Re: The thematic structure was reviewed and changed in introduction, as mentioned before. In the Discussion section, we tried to clarify the comparison with studies that have utilized similar methods or focused on populations in other contexts.
- Methods: The methods section is detailed and provides a clear description of the data sources, sample selection, and analytical techniques. However, there are several areas where additional clarification is needed:
- The rationale for selecting 2006 and 2016 as the comparison years should be more explicitly stated. What specific events or policy changes during this period make these years particularly relevant for studying health inequalities?
Re: As has been stated in the Introduction section we were interested in comparing 2 periods (before and after the financial recession). Moreover, we used the surveys available for these periods. For this reason, we compared these 2 years.
Changes have been made to the introduction and wording of the Work objective to better clarify this point.
- The description of the Primary Care Assessment Tool (PCAT) index is thorough, but the manuscript would benefit from a brief discussion of its validity and reliability, particularly in the Spanish context. Have there been any critiques or limitations noted in previous studies that utilized the PCAT in Spain?
Re: A mention to validity and reliability of PCAT was added in the discussion, as well as other Spanish studies that utilized the PCAT (p. 12, lines 325-331).
- The choice of statistical methods, particularly the use of Somers' D index, should be justified more clearly. Why was this method selected over other potential methods for analyzing inequality patterns?
Re: We have introduced more explanation about the Somers’s D index. See the new text in the Methods section where the test is now better explained. Lines 146-166.
- Results: The results are presented in a detailed manner, but the presentation could be streamlined for better readability. The authors should consider using more visual aids (e.g., graphs or charts) to summarize key findings, especially in the comparison of 2006 and 2016 data. Additionally, the discussion of the results could be more analytical, linking the findings back to the research questions and the broader literature on health inequalities. It is also important to address any limitations in the data or the methodology that might affect the interpretation of the results.
Re: The explanation of the results (Table 4) has been simplified. Also, we have reviewed all the analysis and changed the format of the Tables.
We consider that the coherence of the paper has improved with the changes introduced in the objectives, the introduction and presentation of the results in the modified tables, as well as simplifying the writing of results.
Regarding replacing some table results with a graph, we think that in any case it could be table 3. As we doubt that the graphs are clearer than the table, we include them in this answer, so that you can evaluate it.
- Policy Recommendations: The policy recommendations provided are relevant and grounded in the study's findings. However, they are somewhat generic and could benefit from more specificity. The authors should consider providing more concrete examples of policy interventions that could be implemented to address the identified inequalities. Additionally, the discussion could be expanded to consider the broader implications of the findings for other regions or countries with similar healthcare systems.
Re: Some policy recommendations have been added at the end of the article, along with the conclusions. (p. 13)
Reviewer 3 Report
Comments and Suggestions for Authors
1. is there any explanation for the rather late conduct of the study on secondary data - as much as 8 years after the contractual date of cessation of the economic crisis that also affected public health care in Spain between 2006 and 2016? It can, of course, be assumed that the global economic crisis has hitherto been poorly researched and described in relation to healthcare and the inequalities therein for Spain, and now a convenient opportunity has arisen to make up for this; indeed, the economic crisis was superimposed on the migration crisis, and further on, COVID-19 (which also constituted a crisis); in 1986 Spain joined the EU, in 1975 the dictatorship ended and the transition to democracy began, and the struggle against the economic crisis caused by the years of dictatorship; perhaps a few words about all these crises would have been useful in the introduction, the reader would have had more clarity? It can also be assumed that something meaningful can be said about powerful long-term socio-economic phenomena only after a certain period of time has passed, e.g. when it is known with certainty that they have come to an end;
2. verses 75 d 85 should probably form the 'Aims/Objectives' section.
3. the earlier lines should be a more coherent introduction to the crises and reform of the health sector in Spain, foregrounding the global economic crisis and its intensity in Spain between 2006 and 2016. In essence, the manuscript says very little about the scale and severity of the economic crisis in Spain and how strongly healthcare was affected: what were the cuts and reductions in the offer of treatment services? And on the patients' side - what were the parameters of the crisis? Data on the situation as a whole would help to put the study in a more precisely presented and documented, socio-economic context.
4. In the 'Conclusion' section - firstly - further predictions on the direction of future changes as to the levelling of inequalities should be made clearer - what does this trend specifically depend on (predictors) and what does it mean in detail that 'healthcare has an important role to play here'?; it will soon be another decade since 2016. What is the current situation (regression? crisis?) in Spain and how might this affect healthcare? And secondly, the authors mention inequalities regarding migrants; for more than a decade these inequalities seem to have progressed across Europe, but in Spain and Italy perhaps more than elsewhere. What is now the percentage of foreign-born citizens in Spain as a proportion of the total population? How strong is classism in Spain in terms of access to healthcare? Are the authors talking about public health care throughout? What is the private healthcare sector like and does it have a role in exacerbating the inequalities examined by the authors? Could the authors cite any current research on this topic and also emphasise this research trend in the Conclusion?
Author Response
Response to the Reviewers
We are grateful for the detailed peer review of our work and the particularly timely feedback we have received regarding on our manuscript ‘A Decade of Monitoring Primary Healthcare Functions Through the Lens of Inequality’ [healthcare-3174086], submitted for the Special Issue ‘Perspectives on Primary and Community Healthcare’ of Heathcare. We are also very grateful to the editorial team that has accepted that our work may have some chance of being published in the journal, as well as to the interest shown in the good editorial work.
We have addressed all the comments made and tried to make all the changes suggested for accuracy and quality. In our responses (Re) to each comment, we indicate page (p.) and paragraph (para.) and or lines and modifications with change tracking. We mark the comments received in red and our responses in black and blue.
REVIEWER 3
Thank you very much for the review, the suggestions and improvements that have been proposed. The introduction has been the section that has been modified the most, to be able to meet the suggested proposals, better focusing the work and updating part of the bibliography. In the comments on each of the points indicated in the review, the changes made are specified
The conclusions section has also been modified to a large extent, better identifying what are recommendations, which are proposed based on the results of the study, as well as the existing bibliography and the proposals of the authors of the paper.
Comments and Suggestions for Authors
- is there any explanation for the rather late conduct of the study on secondary data - as much as 8 years after the contractual date of cessation of the economic crisis that also affected public health care in Spain between 2006 and 2016? It can, of course, be assumed that the global economic crisis has hitherto been poorly researched and described in relation to healthcare and the inequalities therein for Spain, and now a convenient opportunity has arisen to make up for this; indeed, the economic crisis was superimposed on the migration crisis, and further on, COVID-19 (which also constituted a crisis); in 1986 Spain joined the EU, in 1975 the dictatorship ended and the transition to democracy began, and the struggle against the economic crisis caused by the years of dictatorship; perhaps a few words about all these crises would have been useful in the introduction, the reader would have had more clarity? It can also be assumed that something meaningful can be said about powerful long-term socio-economic phenomena only after a certain period of time has passed, e.g. when it is known with certainty that they have come to an end;
Re: Reviewing the literature produced in the field of public health and health services on the impact that the economic crisis had on the care provided by the health system, and more specifically primary care, it was found that the population's assessment of their experience at this important level of care was lacking.
The limitation of space in this type of publication we think that does not allow us to delve into important topics such as those pointed out by the reviewer (political system, health system, population and social changes that have occurred in recent decades, etc.). Topics, which, with the exception of the political system, are pointed out in the work; related to the political system, the existence of a national health system is pointed out. We understand the great importance that all these structural determinants have on health and on public policies.
In order to provide a concrete response to this point, we have improved the focus of the work on the economic crisis and on knowing whether the population's assessment of the first level of health care had changed, and if so, what was happening with respect to equity. And we have included two new bibliographic citations, especially that of López-Valcárcel and P Barbe refence num. 14), further broadening the focus of the economic crisis on the health system.
The rationale for the choice of the study period has also been improved, both in the introduction and in the objective of the paper itself.
The economic recession lasted several years and did not end in 2010. In fact, the majority of the cuts started that year, but austerity measures lasted several years, the health expenses didn’t increase until 2015. Reference 15 has detailed indicators.
- verses 75 d 85 should probably form the 'Aims/Objectives' section.
Re: Rather than including the phrases indicated in the objective, we have clarified it, in order to improve coherence with all the results presented, including those related to the assessment of PC according to the characteristics of the doctor of reference. The lack of coherence that existed in the previous version, already pointed out by the reviewers, contributed to the confusion.
the earlier lines should be a more coherent introduction to the crises and reform of the health sector in Spain, foregrounding the global economic crisis and its intensity in Spain between 2006 and 2016. In essence, the manuscript says very little about the scale and severity of the economic crisis in Spain and how strongly healthcare was affected: what were the cuts and reductions in the offer of treatment services? And on the patients' side - what were the parameters of the crisis? Data on the situation as a whole would help to put the study in a more precisely presented and documented, socio-economic context.
Re: As we mentioned previously, the introduction was modified to cover the issues suggested by the reviewer. In the introduction, lines 63-76 have been added, with two new bibliographic references (14 and 15), which help to focus the research question based on the economic crisis of 2008 and specifically whether the population's assessment of their experiences with the first level of health care had changes between two years, 2006 and 2016. The health cuts made in Spain as a result of the economic crisis were not recovered until after 2014. Then, the rationale for comparing 2006 and 2016, and the need to focus on social inequalities has been more clearly established. One of the new bibliographic references included delves into the aspects of the crisis that the reviewer requests (reference 14).
In the 'Conclusion' section - firstly - further predictions on the direction of future changes as to the levelling of inequalities should be made clearer - what does this trend specifically depend on (predictors) and what does it mean in detail that 'healthcare has an important role to play here'?; it will soon be another decade since 2016. What is the current situation (regression? crisis?) in Spain and how might this affect healthcare? And secondly, the authors mention inequalities regarding migrants; for more than a decade these inequalities seem to have progressed across Europe, but in Spain and Italy perhaps more than elsewhere. What is now the percentage of foreign-born citizens in Spain as a proportion of the total population? How strong is classism in Spain in terms of access to healthcare? Are the authors talking about public health care throughout? What is the private healthcare sector like and does it have a role in exacerbating the inequalities examined by the authors? Could the authors cite any current research on this topic and also emphasise this research trend in the Conclusion?
We really appreciate the reviewer's reflections, which encourage us to broaden the discussion and delve deeper into numerous aspects, although we have had to limit ourselves due to editorial restrictions. We focus on public health care throughout because this analysis was conducted at the city’s Agency of Public Health, and our department is the responsible on health statistics there, linked with the health plan and community programs that are designed to reduce social gaps in access to health, and primary care. In particular, migratory movements have been an important issue in Barcelona since the beginning of this century, and numerous actions have been designed to analyze this phenomenon. In this study, immigration has been treated as a homogeneous whole, due to limitations already mentioned. The health survey is a useful approach, but it requires further research with other methodologies that allow us to better understand the possible barriers that health care may entail. Health care is subject to inequality, and therefore it is important that evaluation and preventive strategies include an intersectional approach, taking into account the different axes of inequality. We tried to approach some of the issues proposed by the reviewer in the Discussion section. Please, see paragraphs in pages 11 and 12.
Round 2
Reviewer 1 Report
Comments and Suggestions for Authors
Thanks for your revisions
From table 3 it is still unclear what the P value is for? Comparison of mean or median?
You state "Take into account that many p-values are non significant." but the point about over 90 p values in 1 table increases the chance that the significant p values occurred by chance (type 1 error) It still remains unclear what do these multiple associations mean. Do these test tell us if there was a significant increase or decrease or do they tell us that there were significant differences?
Conclusions appear unnecessarily long and include 2 references. Conclusions should be solely based on the work of the authors without need to cite work of others
Comments on the Quality of English Language345 Moreover, it is important to take into account that the sample size affects the p-values, therefore some small differences could have a small p value, which is why we have described mainly the important changes found in the PC index score.
reword above, suggestion: "it is important to consider that large sample sizes can influence p-values; as a result, minor differences might show statistically significant p-values."
Author Response
Response to the editorial committee:
We are very grateful, both to the editorial committee and to the reviewers, for their work and the promptness in their response. Below we respond to all the comments received, the clarifications that have been requested from us and the responses to some of the proposals that have not resulted in changes.
In the manuscript, the changes made after the first revision have been marked in yellow, and those made because of this second revision have been marked in green.
We request exceptional authorization to make a change in the title of the article, replacing "functions" with "experiences", because it is the correct term, in accordance with the theoretical framework and the measurement performed by the instrument we use (PCAT). We apologize for not having noticed this error in the translation before. However, we have verified that throughout the manuscript we repeatedly use the term "experiences", which is correct. In addition to the title, we changed this word in lines 92 and 293 of the text.
Reviewer 1
Thank you again for your review, which helps us to make the work more understandable.
We will now respond to the points you have raised.
- From table 3 it is still unclear what the P value is for? Comparison of mean or median?
The p value corresponds to the comparison of the overall distributions, not to the comparison of the summary point measures of such distributions. This is because the Distribution of the PC index is not normal, and the samples are weighted. If they were not weighted, the Mann-Whitney U test would have been used.
Often, the test result coincides with the differences observed in the medians, but not always, and in some cases equal or very similar medians correspond to different distributions because extreme values may influence the test and not be reflected in the medians.
Thus, for example, it can be seen in table 3 that the median in women is the same both years (73.3), in the following figure you can see a change in the distribution of the index between both years, this being the cause of the p-value being significant:
To make it clearer what the p-value corresponds to, we have modified the title of table 3, as well as that of table 2. And changes have been made in the Methods section, to explain it more clearly (lines 150-154). An explanatory sentence has also been included in the results section of Table 3 (lines 194-196)
- You state "Take into account that many p-values are non significant." but the point about over 90 p values in 1 table increases the chance that the significant p values occurred by chance (type 1 error) It still remains unclear what do these multiple associations mean. Do these test tell us if there was a significant increase or decrease or do they tell us that there were significant differences?
The comparison of the distributions with the Summers' D test is bilateral. The hypothesis about the difference that the test contrasts is bilateral, about whether there are differences, not the direction of the differences. Observation of the distributions (not shown) and estimates of the summary measures (mean, median and interquartile range) help to interpret the direction of the changes.
- Conclusions appear unnecessarily long and include 2 references. Conclusions should be solely based on the work of the authors without need to cite work of others
Certainly, this section includes conclusions and recommendations. In the first review, Reviewer 2 asked us to expand this part, with greater specificity of the recommendations and we considered it appropriate. Since Reviewer 2 has not commented on this section, we have decided not to modify it.
The group of authors believes that a public health research paper should also conclude by recommending what the conclusions suggest to move forward.
But understanding that he is right in that the conclusions are from the work and not from bibliographic references, we have included in the title of the section "and recommendations", waiting for the editorial committee to agree with this proposal.
- Moreover, it is important to take into account that the sample size affects the p-values, therefore some small differences could have a small p value, which is why we have described mainly the important changes found in the PC index score.
reword above, suggestion: "it is important to consider that large sample sizes can influence p-values; as a result, minor differences might show statistically significant p-values."
We believe that the proposal is very appropriate. It has been done. (lines 328-330)

Reviewer 2 Report
Comments and Suggestions for Authors
This paper has been improved a lot, but this literature should be cited
10.1177/00469580241266402
Author Response
Response to the editorial committee:
We are very grateful, both to the editorial committee and to the reviewers, for their work and the promptness in their response. Below we respond to all the comments received, the clarifications that have been requested from us and the responses to some of the proposals that have not resulted in changes.
In the manuscript, the changes made after the first revision have been marked in yellow, and those made because of this second revision have been marked in green.
We request exceptional authorization to make a change in the title of the article, replacing "functions" with "experiences", because it is the correct term, in accordance with the theoretical framework and the measurement performed by the instrument we use (PCAT). We apologize for not having noticed this error in the translation before. However, we have verified that throughout the manuscript we repeatedly use the term "experiences", which is correct. In addition to the title, we changed this word in lines 92 and 293 of the text.
Reviewer 2
This paper has been improved a lot, but this literature should be cited: 10.1177/00469580241266402
We thank reviewer 2 for his prompt review and positive assessment of the changes made, which we thank to the suggestions of the three reviewers of the first round of the review.
In this case, the reviewer proposes to add a new reference to a good article on the impact of the drastic public health measures that the COVID-19 pandemic caused (strict lockdown) on the paralysis of supplies to a territory, isolating it from the environment. It describes the consequences on the economic growth of the territory, through:
- Decrease in supplies
- Decrease in demand
- Increased costs due to increases in transport
- Hindering the flow of capital
- Decreases in profits expected by companies
After discussion, the authors considered that, given that it refers to a very different type of crisis and the study of a specific factor, despite its high relevance, also taking into account that we have not included the global crisis of the pandemic of COVID-19, and that we had expanded the bibliography and the work in reference to the economic crisis of 2008, we have not included it. We kindly ask the reviewer to accept this decision.
